# Space Analysis of School Violence in the Educational Setting of Peru, 2019

**DOI:** 10.3390/ijerph192316044

**Published:** 2022-11-30

**Authors:** Wendy Arhuis-Inca, Miguel Ipanaqué-Zapata, Janina Bazalar-Palacios, Jorge Gaete

**Affiliations:** 1Universidad Señor de Sipán, Chiclayo 14001, Perú; 2Vicerrectorado de Investigación, Universidad Privada Norbert Wiener, Lima 15001, Perú; 3Universidad Tecnológica del Perú, Lima 15001, Perú; 4Faculty of Education, Research Center for Students Mental Health (ISME), Universidad de Los Andes, Santiago 1030000, Chile; 5National Research and Development Agency (ANID), Millennium Science Initiative Program, Millennium Nucleus to Improve the Mental Health of Adolescents and Youths, Imhay, Santiago 1030000, Chile

**Keywords:** school violence, students, administrative staff, spatial analysis

## Abstract

Background: Schools are increasingly experiencing physical, psychological, and sexual violence, which impacts students’ academic achievement and physical and emotional health. Our objective was to identify regional prevalence rates, average prevalence by aggressor type, and provincial spatial conglomerates with higher rates for each type of school violence reported in educational settings in Peru during 2019. Methods: An analysis was made of 12,132 cases reported through Peru’s Specialized School Violence System (SíseVe). The Moran indices were calculated using provincial prevalence rates to identify conglomerates with high prevalence. Results: In the coast region, the Department of Tacna reported the highest prevalence rates for physical (99.7) and psychological (107); the Department of Amazonas, which is in the jungle, reported the highest prevalence rate for sexual violence (74.6), with teachers in public schools accounting for the majority of sexual assaults against students (56%). Conclusions: Sexual violence predominated in the jungle zone, with a significant percentage of teachers participating in violence with sexual connotations. Physical and psychological violence prevailed in the coastal region.

## 1. Introduction

School violence is any type of physical, psychological, or sexual aggression directed at or between children and adolescents in educational settings. The perpetrator could be a fellow student, a teacher, or another school staff member [1]. One in every two people between the ages of 2 and 17 worldwide suffers from psychological, physical, or sexual abuse [2]. Violence against this study population is a public health issue that has negative consequences for physical and mental health [2,3] and significant economic costs [4]. This violence also results in a breach of rights, as acknowledged by several international organizations [5,6,7].

There is little evidence of the problem of violence among students, but few studies of violence perpetrated by teachers or other administrative school staff worldwide and in Latin America [1,8]. According to the 2019 UNESCO report, kids have been physically and psychologically abused for a long time, and 68 countries still allow corporal punishment [1]. Research in low- and middle-income nations, where sexual assault is prevalent, is scarce [9]. Among these few studies, we can mention one conducted in Malawi, where 15% of girls and 19% of boys reported having been sexually assaulted by a teacher or a peer at school [10]. In Liberia, it was found that 30% of girls were victims of sexual assault by religious and educational professionals [11]. This evidence is essential to trigger regulatory adjustments for the student’s benefit.

Being a victim or an offender of school violence has been linked to various individual, family, school, and community factors. The features of the offender or victim are influenced by demographic variables, including gender, age, race, immigration status, handicap, and socioeconomic level [12,13,14]. Some of the perpetrator’s characteristics found in the literature are impulsivity, hostility, fear of social rejection, antisocial, and illegal behaviour [15,16]. At the family level, the perpetrator is affected by parental relationships, parental monitoring, and unfavourable parenting [17,18,19]; the latter is also connected to being a victim of violence [13]. Other risk factors for being a victim and perpetrator of violence are school size, school climate, disorder and school safety [20,21,22,23]; and at the community level, crime and economic deprivation [18,24]. The most disadvantaged schools with the highest rates of community delinquency also have high rates of violence, and those living in rural areas or with a low socioeconomic status are prone to violence [12,17,18]. In response to the community and geographic factors associated with school violence, the Geographic Information System (GIS) was developed as an alternative to display school violence hotspots. The GIS analyzes and connects numerous types of data to a single geographic place. To effectively devote resources to prevention efforts, countries like the United States and Brazil have employed this technique to geographically analyze data and identify high-risk locations for school violence [25,26,27,28]. However, none of these studies analyzed the data using a more comprehensive approach.

The construction of educational health policies to decrease or eliminate violence against students in schools may be accomplished with the help of the GIS. It can be used to insert preventative activities, follow up, monitor, and create educational health policies. The applicability of this technique to identify the hotspots for any kind of school violence has not yet been studied in Peru. In particular, few studies have explicitly looked at the violent acts committed by teachers or school staff, especially those who have participated in violence with sexual connotations [29]. For the above, the present study aimed to determine regional prevalence rates, average prevalence by aggressor type, and provincial spatial conglomerates for every kind of school violence reported in educational settings in Peru during 2019.

## 2. Materials and Methods

### 2.1. Study Design and Setting

An ecological study was conducted using cases reported according to types of school violence within Regular Basic Education Institutions (IEBR) in 2019 through the virtual platform Peru’s Specialized System for Reporting Cases of School Violence (SíseVe). It evaluated 12,132 reported cases of school violence through its open access platform (http://www.siseve.pe/Web/ accessed on 27 November 2022). The platform allows for georeferenced information obtained through the Local Management Units of Peru, making it possible to analyze GIS projects. In Peru, there are 220 local educational management units, defined as autonomous provincial institutions in charge of designing, executing, evaluating educational policies and supervising educational services provided by educational programs and institutions according to quality standards [30].

The information was evaluated by grouping the cases at the level of the provinces of Peru, where the educational institutions of Peru are located. Peru is a South American country located west of the Pacific Ocean with a total area of 1,285,216 km^2^ [31]. It is geographically divided into three natural regions (Coast, Highlands and Jungle), consisting of 25 departments and 196 provinces.

### 2.2. Data Sources

The data source of the Specialized System of School Violence (SíseVe) was created by the Peruvian state through the Ministry of Education (MINEDU) in September 2013 with the aim of reporting, managing and addressing cases of school violence within Regular Basic Education Institutions (IEBR) at the national level [32]. This database represents an important source of information due to a large number of reliable and verified real cases, taking into account that the identity document of the reporter is required for registration and is validated by the SíseVe team [33].

The study included georeferenced cases of school violence collected by local educational management units during the last year with presential assistance before the COVID-19 pandemic in Peru (January to December 2019). School violence was classified according to the United Nations Educational, Scientific and Cultural Organization (UNESCO) as physical (physical attacks, physical fighting, corporal punishment and bullying), psychological/verbal (verbal and emotional abuse, social exclusion and psychological harassment) and sexual (teasing, taunting, sexual comments or gestures) [1].

Some databases were progressively added to obtain the final database used in the spatial analysis by provincial spatial clusters of school violence. First, the initial database collected from the SiseVe platform and MINEDU’s National Registry of Local Educational Management [34], were merged to obtain a database identifying the local educational management units at the micro-regional level (departments and provinces). Subsequently, to obtain provincial prevalence rates for 2019, the resulting database was combined with the MINEDU Educational Quality Statistics database [35], which includes the total number of students enrolled per province, to obtain provincial prevalence rates per 100,000 students. After getting the units of analysis (provincial rates) for the spatial analysis, a map of Peru’s 196 provinces was constructed using the shapefile format (shp.) of the National Institute of Statistics and Informatics (INEI) cartographic base.

### 2.3. Data Analysis

To run the descriptive analysis, this study used cross-tabulations of the different types of school violence during 2019 by department, geographical region, and perpetrator types.

The spatial analysis of the types of school violence was conducted in three stages to identify the provincial spatial clusters with a high prevalence rate in 2019. The first stage consisted of integrating the databases with information on the prevalence rates of school violence for each province with the map in shapefile format (shp.), generating a layer for each type of school violence in the 195 identified provinces (Figure 1). Several tools were used to obtain the spatial patterns (i.e., Moran’s indexes, Getis and Ord indexes, among others). For the study, Moran’s indexes (global and local) were taken into account because they are the most widespread and used for their direct interpretation of spatial associations [36]. Next, the first spatial autocorrelation indicator called Moran’s global index (also identified as Moran’s I’) was evaluated to determine the clustering pattern (clustered, dispersed, or random) of the country-level data for the provincial prevalence rates of the types of school violence. The clustered pattern is the spatial distribution where clusters with similar spatial values (intensity level) are generated within the units of analysis. In contrast, the dispersed pattern refers to the spatial distribution of the units of analysis, not generating clusters with similar spatial values but attracting different spatial values. On the other hand, the randomized pattern does not attract any spatial value or form clusters [37]. The interpretation of Moran’s I’ is obtained by contrasting the null hypothesis (Ho), evaluating the presence of randomness in the data with a significance level (α) at 5% [38], which excludes further analysis if the Ho is accepted. This indicator can be interpreted using the Moran’s I’ value or Z-score (a standardized statistical indicator for contrasting hypothesis testing) and *p*-value. Moran’s I’ is between −1 and 1. If the statistical index is greater than 0, there is clustering between data with similar frequencies (high–high or low–low); if it is less than 0, there is dispersion or rejection of data with similar frequencies (high–low or low–high groups); finally, if it is 0, there is randomness (it does not generate any group of values). The other type of interpretation refers to whether the z-score is positive and the *p*-value indicates clustering, whether the z-score is negative and the *p*-value indicates dispersion, and whether the *p*-value indicates randomness [37], with the latter interpretation being the one used in the current study.

After identifying the types of violence that presented clustering, the provincial clusters for each type of school violence were located using Morán’s local index (also called Local Indicator of Spatial Association or LISA). This indicator evaluates its grouping hypothesis with a significance level of 5%, generating three spatial clusters: provinces with similar prevalence rates to their neighbouring provinces (high–high or low–low), provinces with different prevalence rates from their neighbouring provinces (high–low or low–high), and provinces with prevalence rates that are not related to their neighbours (not significant). For our study, it was important to identify the area with the highest prevalence rate (high–high, also called hotspots) for each type of violence in 2019.

To perform these spatial autocorrelation analyses, it is crucial to generate spatial weights since these weights are intended to connect and relate the spatial units of polygonal formation and generate the clusters. In many cases, contiguity-based clustering is used to obtain spatial weights, rendering a pairwise evaluation of polygonal spatial units through an edge, vertex, or common point [39]. The Queen or Tower criterion can be used to evaluate groupings based on contiguity, where the former shows the linkage between spatial units with a common vertex or point. In contrast, the Tower criterion is only linked through a common vertex. The present study generated spatial weights through provincial units using Reina’s contiguity criterion. Moran’s spatial analysis is used with 999 permutations, indicating that spatial units are randomly rearranged 999 times to evaluate the null hypothesis of no spatial clustering [40]. Subsequently, the provinces within this hotspot cluster were classified according to Peruvian departments and urban or rural residential areas. The programs used for descriptive analysis were the statistical software STATA 16 (StataCorp LP, College Station, TX, USA) and for spatial analysis GeoDa 1.14.0 (GeoDa Center for Geospatial Analysis and Computation, Arizona State University, Tempe, AZ, USA).

## 3. Results

A total of 195 provinces were assessed in 2019, with a total of 12,132 cases of school violence in the IEBRs. The mean age of perpetrators in some forms of violence ranged from 10 to 13 years, with sexual violence having the highest mean age (mean = 12.7; Sd = 3.0) and physical violence the lowest (mean = 10.8; Sd = 3.8). Men suffered more physical violence, reporting a prevalence rate of 93.0, and women suffered more psychological and sexual violence with prevalence rates of 59.5 and 47.9, respectively. Regarding natural regions, the highest prevalence rates were found on the coast for physical and psychological violence (81.7 and 60.1), except for sexual violence, which was found in the jungle region (35.6). The department of Tacna reported the highest prevalence rates of physical (99.7) and psychological violence (107), while the highest prevalence rate of sexual violence was reported by the department of Amazonas (74.6) (Table 1).

The teachers were the perpetrators of most violence against students, with the highest incidence of sexual violence (56.0%). In psychological and sexual violence cases, only one instance of aggression was reported in 51.2% and 43.5%, respectively. In most psychological and sexual violence cases, the aggression frequency was reported to be only once, with 51.2% and 43.5%, respectively. In general, the types of school violence are exercised by a single person, especially sexual violence (94.0) (Table 1).

The spatial analysis showed through Moran’s I’ that there was spatial autocorrelation with a tendency to cluster in physical violence with 0.133 (z = 3.28; *p* = 0.002), psychological with 0.038 (z = 4.31; *p* = 0.004) and sexual with 0.070 (z = 1.74; *p* = 0.047). Of the 195 evaluated provinces, the LISA map showed that the number of provinces with higher than average prevalence rates and surrounded by neighbours with the same frequency (hotspots or also called high–high) was 7 for physical violence, 8 for psychological violence, and 11 for sexual violence; all were located in the southern part of Peru for all types of violence, with the northern jungle provinces only being added for sexual violence. It is important to mention that this type of spatial analysis seeks to identify clusters between nearby provinces with high prevalence rates of school violence (high–high); however, clusters can also be generated between nearby provinces with low prevalence rates (low–low) or clusters of provinces with high or low prevalence rates (high–low or low–high); the latter two types of clusters are irrelevant for the identification of hotspots, which is the objective of the study (Figure 2).

The provinces identified within the hotspot cluster reported that most physical violence was found in rural areas (4/7). The hotspot cluster of the prevalence of psychological violence reported that 4 out of 8 provinces were located in urban and rural areas. For sexual violence, it was reported that (9/11) was exercised in urban areas. While it is true that the clusters with the highest prevalence of provinces reporting some type of physical and psychological violence are located in the department of Tacna (Southern Peru), it is important to mention that, in sexual violence, hotspot provinces from the jungle region were included (Amazonas = 2/5; San Martin = 3/8) and the Highlands (Ayacucho = 3/7) (Appendix A).

## 4. Discussion

This is the first study that identifies the provincial spatial conglomerates with high prevalence rates of school violence according to type in educational establishments in Peru during 2019. The prevalence rates of physical and psychological violence were higher in the department of Tacna, located in the coast region, and in the department of Amazonas, located in the jungle region, due to sexual violence. Likewise, according to the type of aggressor, one out of every two cases of sexual violence was caused in the educational establishment and exercised by the teacher. Finally, the spatial analysis reported provincial spatial groupings for the prevalence of sexual violence distributed in the jungle region.

Physical violence is less common than psychological/verbal aggression, consistent with reports from other countries [41]. The prevalence rates of psychological/verbal violence were similar to those reported in Peruvian studies [29,42,43]. According to a multinational investigation of Latin America, Argentina, Peru, Costa Rica, and Uruguay had the highest prevalence rates of psychological/verbal violence among children [44]. The use of the media has been investigated in recent years concerning psychological violence and may be related to an increase in this type of violence towards students and between peers [8]. In many regions of Peru, the prevalence rates of these types of violence vary. However, the explanation adds that Tacna, located on the southern coast of Peru, is a city with marginal urban areas, mainly occupied by migrants, people with few economic opportunities, or where public servants have not been able to meet basic needs because it is a poorly organized community, all of which are risk factors for physical violence [45].

On the other hand, we discovered high rates of sexual violence in the Peruvian Amazon, which belongs to the Selva region. Victims of violence often do not file a complaint because they accept it as a common occurrence due to the number of circumstances they face and the fear of future reprisals [46]. Reporting a violation is a betrayal of the family, an offence to others, or the risk of being left without leaders or teachers, especially in the jungle [47]. As sexual assault has become commonplace, boys are forced to confront these injustices and live in constant fear of seeing their ambitions thwarted [46].

More than half of the acts of sexual violence were perpetrated by teachers of the educational establishment. Although there are studies on violence in educational establishments, most of them are among schoolchildren, and few report the teacher as an aggressor [48]. A limited number of studies corroborate our findings. A literature review in Liberia revealed a high prevalence of sexual violence against students perpetrated by teachers and religious school workers [11]. Similarly, a cohort study in Peru identified a higher prevalence of psychological violence perpetuated against female students by their teachers [41]. In this study, we include the type of aggressor in our analysis, given the importance of knowledge and far from the implications they could have for the functioning of the educational establishment. However, contrary to Law 29944 “Teaching Reform,” which accounts for acts of violence in educational establishments [49] and does not specify punishment or monitoring of the teacher who commits the act of violence, the teacher is frequently relocated to another school, without major consequences in the sector. For this reason, it is imperative to make regulatory adjustments to improve the corresponding sanctions.

The spatial analysis of provincial groups shows a clear pattern of the distribution of sexual violence in the jungle region. According to data from the Defensoria del Pueblo, between 2012 and 2017, 216 complaints were filed with the Unidad de Gestión Educativa Local of a city in the jungle of Peru for sexual violence in schools [50]. Likewise, according to the National Police of Peru, 6 out of 10 cases out of the total number of complaints of sexual violence registered correspond to girls and adolescent women between 7 and 17 years of age. In 2016, the National Police of Peru registered 3683 reports of rape from this population group, an average of 10 cases per day [51]. The vulnerability of girls and adolescents in the jungle exposes them to violence, even at school. Sexual violence affects physical and mental health and violates the fundamental rights of this population [52]. One consequence of this problem is high indicators of adolescent pregnancy, even unwanted pregnancies, in these regions, at double the national average [53], as well as school dropouts [52], and the subsequent impact on the development of the life plans of girls and adolescents from these communities [54,55]. This makes visible the inequalities and exclusion that persist in Peru, especially in the Amazonian communities. There are barriers that girls and boys from rural communities in the Amazon must face regarding access the SiseVe platform or other justice, protection, and welfare services, which are still unknown to the authorities of the educational sector in Peru.

Given our findings, it is necessary to carry out more research in the provinces and regions that report the most school violence for political decision-making at the national level. In 2011, Law 29719 was declared—Law to Promote Coexistence without Violence in Schools [56]—represents progress with the opening of the Specialized School Violence System (SiseVe, for your acronym in Spanish) in 2013, which received case reports of school violence at the national level from the victim or witness, and also presents modules on how to prevent violence. However, direct intervention should be used to reduce the possibility of escalation to severe, or even fatal, consequences. These interventions may help to guarantee inclusive, equitable, and quality education and promote lifelong learning opportunities for all, with safe, non-violent, inclusive, and effective learning environments for all [57]. For example, preventive measures to reduce school violence may include social and emotional learning interventions to improve the school climate or reduce blind spots in the school premises, increasing adult supervision to improve school safety. These measures have been used in several evidence-based programs, such as the Olweus Program [58] or KiVa program [59].

The main strengths of the study are the following: the use of types of school violence according to the definition standardized by UNESCO; the information from the national level is accurate and reliable; and this is the first study of school violence, showing results through the geographic information system (GIS), to visualize the spatial behaviour of the prevalence rates of school violence. This will make it possible to focus resources and implement interventions for the prevention, monitoring, and the creation of health and educational policies to improve the situation regarding violence in schools against students. Among the limitations of the study, information bias is latent since there may still be low reporting of acts of violence suffered by students; furthermore, there may be a limitation in the representation of the sample and potential report bias because students may underreport incidents of violence if they fear the consequences or cultural stigma. However, in 2018, the reporting of school violence began to be promoted in a mandatory manner by students through a Peruvian government-approved supreme decree (DS N°004–2018–MINEDU) [60]. Another limitation of the study was that we did not have access to reliable information about several potential differences between regions to explore the results of our study. Further research is needed.

Likewise, the barriers that girls and boys from rural communities in the Amazon must face in accessing the SíseVe platform, and the gap in reported cases between the Unidad de Gestion Educativa Local in urban and rural areas may indicate a reporting bias. However, the results are similar to previous studies on school violence [52,53].

## 5. Conclusions

In conclusion, physical and psychological violence predominated in the coastal region and sexual violence in the jungle region. According to the aggressor, the teacher was responsible for half the cases of sexual violence in the educational establishment. Finally, the spatial analysis reported provincial spatial groupings for the prevalence of sexual violence distributed in the jungle region. Future research should concentrate on identifying the social factors and determinants that may explain the geographic differences in Peruvian school violence, for example, to explore socio-economic and demographic factors, cultural and family differences among others. To explore these potential explanations, we suggest including qualitative methods as a first approach, especially regarding the understanding of cultural issues.

## Figures and Tables

**Figure 1 ijerph-19-16044-f001:**
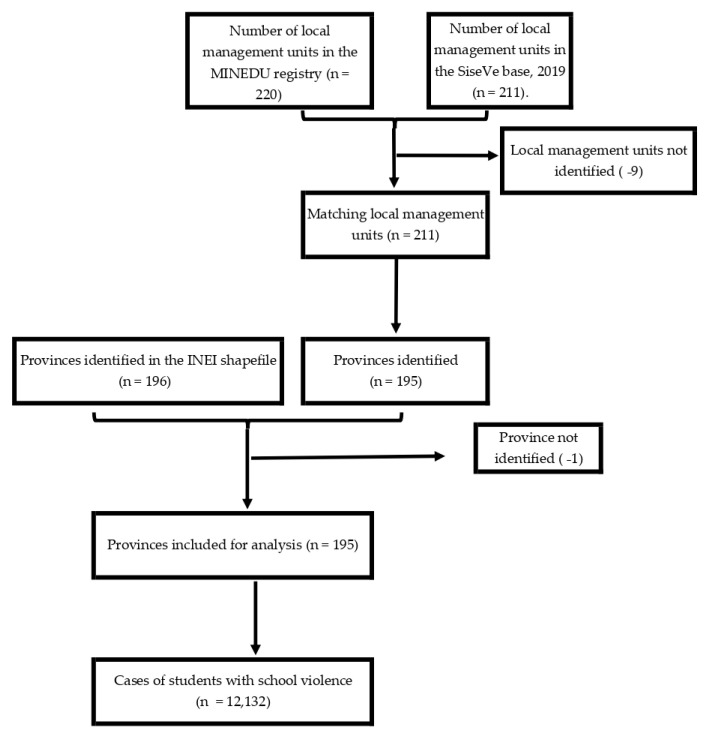
Development of the provincial cartographic base of school violence in the IEBR, 2019.

**Figure 2 ijerph-19-16044-f002:**
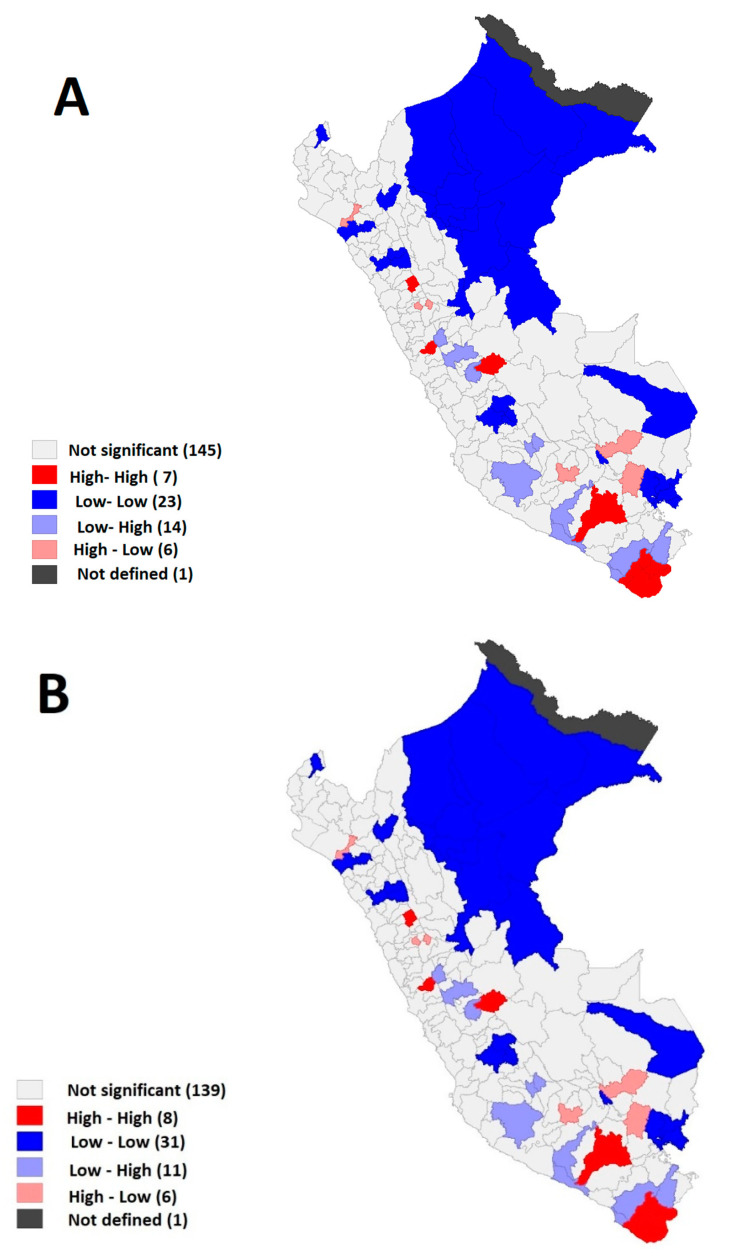
(**A**). Spatial analysis of provincial prevalence rates of physical violence in students of traditional primary education institutions in Peru in 2019. (**B**). Spatial analysis of provincial prevalence rates of psychological violence in students of traditional primary education institutions in Peru in 2019. (**C**). Spatial analysis of the provincial prevalence of sexual violence in students of traditional primary education institutions in Peru in 2019.

**Table 1 ijerph-19-16044-t001:** Characteristics of the study sample according to types of violence.

Variables	Physical (n = 5841)	Psychological (n = 4104)	Sexual (n = 2187)
Age			
Mean (Sd)	10.8 (3.8)	11.7 (3.5)	12.7 (3.0)
Sex of the victim ^a^			
Male	3793 (93.0)	1758 (43.1)	298 (7.3)
Woman	2048 (51.9)	2346 (59.5)	1889 (47.9)
Educational level ^a^			
Initial	663 (37.5)	328 (18.5)	59 (3.3)
Elementary	2358 (64.3)	1426 (38.9)	611 (16.7)
Secondary	2820 (108.9)	2350 (90.8)	1517 (58.6)
Natural Region ^a^			
Coast	4475 (81.7)	3275 (60.1)	1431 (23.4)
Highlands	971 (59.8)	659 (41.9)	508 (28.2)
Jungle	395 (48.8)	170 (22.3)	248 (35.6)
Area de residence ^a^			
Urban	5023 (73.8)	3607 (53.0)	698 (21.9)
Rural	818 (67.2)	497 (40.9)	1489 (57.4)
Departments ^a^			
Amazonas	83 (61.3)	48 (35.5)	101 (74.6)
Ancash	233 (79.7)	134 (45.8)	78 (26.7)
Apurímac	35 (28.8)	27 (22.2)	29 (23.9)
Arequipa	284 (83.7)	264 (77.8)	99 (29.2)
Ayacucho	125 (69.3)	98 (54.4)	66 (36.6)
Cajamarca	173 (42.6)	113 (27.9)	143 (35.2)
Callao	196 (79.8)	187 (76.1)	61 (24.8)
Cusco	166 (48.3)	124 (36.1)	67 (19.6)
Huancavelica	41 (38.2)	12 (11.2)	38 (35.4)
Huánuco	185 (85.5)	90 (41.6)	81 (37.4)
Ica	215 (92.0)	149 (63.7)	48 (20.5)
Junín	264 (78.1)	172 (50.9)	96 (28.4)
LaLibertad	274 (56.0)	180 (36.8)	115 (23.5)
Lambayeque	150 (46.4)	101 (31.2)	47 (14.5)
Lima ^1^	2277 (89.3)	1774 (69.6)	575 (22.6)
Loreto	66 (18.7)	29 (8.2)	93 (26.3)
MadredeDios	48 (94.0)	26 (50.9)	10 (19.6)
Moquegua	27 (62.8)	27 (62.8)	20 (46.5)
Pasco	55 (74.9)	31 (42.2)	19 (25.9)
Piura	469 (85.8)	226 (41.3)	160 (29.3)
Puno	58 (19.9)	69 (23.7)	49 (16.8)
SanMartín	220 (85.6)	83 (32.3)	91 (35.4)
Tacna	82 (99.7)	88 (107.0)	22 (26.8)
Tumbes	54 (79.3)	20 (29.4)	25 (36.7)
Ucayali	61 (33.5)	32 (17.6)	54 (29.6)
Type of perpetrator ^2^			
Student	3890 (66.6)	1552 (37.8)	650 (29.8)
Assistant	106 (1.8)	123 (3.0)	95 (4.4)
Director	83 (1.4)	207 (5.1)	40 (1.8)
Teacher	1700 (29.1)	2126 (51.8)	1223 (56.0)
Support Staff	51 (0.9)	90 (2.1)	132 (6.0)
Staff from other IE	9 (0.2)	5 (0.1)	44 (2.0)
Sex of the perpetrator ^2^			
Male	3697 (63.3)	1771 (43.2)	2073 (94.9)
Woman	2142 (36.9)	2332 (56.8)	111 (5.1)
Frequency of aggression ^2^			
Once	4039 (69.2)	1676 (40.8)	1235 (56.5)
2 to 3 times	1223 (20.9)	1288 (31.4)	642 (29.3)
4 to 5 times	230 (4.0)	420 (10.2)	131 (6.0)
6 times or more	349 (5.9)	720 (17.6)	179 (8.2)
Number of perpetrators ^2^			
1	5220 (89.4)	4104 (87.2)	2055 (94.0)
2 or more	621 (10.6)	526 (12.8)	132 (6.0)

^1^ Includes Metropolitan Lima provinces. ^2^ Variables presented in percentages (%). ^a^ Prevalence rate per 100,000 students within the EBR, 2019.

## Data Availability

http://www.siseve.pe/Web/.

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
