# Peer review of "Space Analysis of School Violence in the Educational Setting of Peru, 2019"

_ijerph, 2022, doi:10.3390/ijerph192316044_

Round 1

Reviewer 1 Report

 Authors discussed very important topic, namely school violence at or between children and adolescents in educational settings, which requires ongoing monitoring, analysis and effective prevention. Main goal of the research is clearly indicated in the manuscript. Authors presented the obtained results in an understandable manner, also with the use of tables and figures (statistical analysis, spatial diversification of the results - spatial analysis of provincial prevalence rates of school violence), especially the figure 2 is helpful for readers with spatial diversification (minor remark below), and, additionally, in the discussion Authors shed more light on the prevalence rates of various types of school violence and provided some explanations related to schools violence (with sources of data and information, the references are appropriate).   

Conclusions are consistent with the evidence and arguments presented, and the conclusions address the main goal of the research - ‚Our objective was to identify regional prevalence rates, average prevalence by aggressor type, and provincial spatial conglomerates with higher rates for each type of school violence reported in educational settings in Peru during 2019’. However, my suggestion relates to a future research section (e.g. as Conclusions and Future Research) - Authors can consider to add it and indicate more detailed areas and topics for further research (methods - maybe qualitative methods, e.g. semi-structured interviews, and research questions, content analysis of media, e.g. local and national media, social media, other indicators, or other examples chosen by Authors). Authors mentioned that „future research should focus on determining the social factors and determinants that may explain these geographical differences’, so more details on a future research related to these social factors and determinants can be given here. This part of the manuscript will increase contribution to scholarship and academic soundness.   

The topic is relevant as well as Authors' contribution to the research within school violence. The results obtained by Authors are based on combination of both statistic analysis and spatial analysis of school violence, and as Authors indicated - 'it is the first study of school violence showing results through the geographic information system (GIS), to visualize the spatial behaviour of the prevalence rates of school violence’. Authors provided a kind of framework to this type of analysis.   

Study design and setting, data sources, and data analysis were presented and discussed clearly and in an understandable manner. Research procedure within spatial analysis was divided in three stages and each stage was discussed separately by Authors in the material and methods, and the figure 1 is also helpful to understand the research procedure. Please check the numbering: 2.2. Data sources, 2.3. Data sources.   

However, my knowledge within spatial analysis, statistics and statistical analysis is limited and I don't feel qualified to judge about these parts of the manuscript (Is the research design appropriate? Are the methods adequately described?) - I mean, e.g. assumptions of spatial autocorrelation analyses or correctness of the results, the interpretation of Moran's I’. The results are presented rather clearly and in an understandable manner, however Authors can consider to add some additional explanations to figure 2 with provincial prevalence rates of school violence, what does precisely mean the scale - high-high, low-low, low-high etc.?

Author Response

Respuesta al revisor 1 Comentarios

Punto 1: Los autores discutieron un tema muy importante, a saber, la violencia escolar en o entre niños y adolescentes en entornos educativos, que requiere un seguimiento, análisis y prevención efectivos permanentes. El objetivo principal de la investigación está claramente indicado en el manuscrito. Los autores presentaron los resultados obtenidos de manera comprensible, también con el uso de tablas y figuras (análisis estadístico, diversificación espacial de los resultados - análisis espacial de las tasas provinciales de prevalencia de violencia escolar), especialmente la figura 2 es útil para lectores con diversificación espacial (comentario menor a continuación) y, además, en la discusión, los autores arrojaron más luz sobre las tasas de prevalencia de varios tipos de violencia escolar y brindaron algunas explicaciones relacionadas con la violencia escolar (con fuentes de datos e información,

Response 1: Ok, thanks.

Point 2: Conclusions are consistent with the evidence and arguments presented, and the conclusions address the main goal of the research - ‚Our objective was to identify regional prevalence rates, average prevalence by aggressor type, and provincial spatial conglomerates with higher rates for each type of school violence reported in educational settings in Peru during 2019’. However, my suggestion relates to a future research section (e.g. as Conclusions and Future Research) - Authors can consider to add it and indicate more detailed areas and topics for further research (methods - maybe qualitative methods, e.g. semi-structured interviews, and research questions, content analysis of media, e.g. local and national media, social media, other indicators, or other examples chosen by Authors). Authors mentioned that „future research should focus on determining the social factors and determinants that may explain these geographical differences’, so more details on a future research related to these social factors and determinants can be given here. This part of the manuscript will increase contribution to scholarship and academic soundness.

Response 2: Thank you for your advice. We have added the following paragraph: “Future research should concentrate on identifying social factors and determinants that may explain the geographic differences in Peruvian school violence. For example, to explore socio-economic and demographic factors, cultural and family differences among others. To explore these potential explanations, we suggest to include qualitative methods as a first approach, especially regarding understanding cultural issues”. We consider that the subheading “Conclusions” is still suitable for this section; therefore, we suggest to keeping it as it is now. See lines 336-341.

Point 3: The topic is relevant as well as Authors' contribution to the research within school violence. The results obtained by Authors are based on combination of both statistic analysis and spatial analysis of school violence, and as Authors indicated - 'it is the first study of school violence showing results through the geographic information system (GIS), to visualize the spatial behaviour of the prevalence rates of school violence’. Authors provided a kind of framework to this type of analysis.

Response 3: Ok, tkanks.

Point 4: Study design and setting, data sources, and data analysis were presented and discussed clearly and in an understandable manner. Research procedure within spatial analysis was divided in three stages and each stage was discussed separately by Authors in the material and methods, and the figure 1 is also helpful to understand the research procedure. Please check the numbering: 2.2. Data sources, 2.3. Data sources.   

Response 4: We have edited and corrected the numbering as suggested.

Point 5: However, my knowledge within spatial analysis, statistics and statistical analysis is limited and I don't feel qualified to judge about these parts of the manuscript (Is the research design appropriate? Are the methods adequately described?) - I mean, e.g. assumptions of spatial autocorrelation analyses or correctness of the results, the interpretation of Moran's I’. The results are presented rather clearly and in an understandable manner, however Authors can consider to add some additional explanations to figure 2 with provincial prevalence rates of school violence, what does precisely mean the scale - high-high, low-low, low-high etc.?

Respuesta 5: Apreciamos su comentario y se incluyó en el texto una nueva explicación de la escala (alto-alto, bajo-bajo, bajo-alto, alto-bajo) utilizada en los análisis (líneas 207–215). Estas explicaciones ayudan a comprender mejor la Figura 2.

Reviewer 2 Report

-This is a paper with a very interesting topic of study: the different types of violence and its prevalence. It emphasizes how important is to be aware of the scope of this problem and how it influences on different populations, especially children.

I think it is relevant as I believe the more information we get about a problematic the more possibilities we have to tackle it.

-Besides, the authors get interesting results with clear differences according to regions in the types of more common violence.

-Nevertheless it seems more a sociological study than a psychological study. It talks about prevalence but the paper doesn’t mention if the authors analyze the characteristics of all regions to explain the differences and its relevance.

Moreover, the paper doesn’t deep on possible measures and tools to address the issue. Authors point to the variables that could increase the prevalence of violence as school size, school climate, disorder and school safety, etc. But they don’t delve into justification to its full extent.

-Furthermore, regarding the sample I find it difficult to assure reliability with the sample and cases of violence.

In this sense, authors recognize that the sample could be bias because of the cases in rural areas, with more difficulties to access to the platform to report the information about their situation. Nevertheless, they don’t explain to what extent they can ensure they got most of the cases of violence and how they controlled the process.

-Finally, in the conclusions authors only focus on the data they got but they didn’t deep on measures to approach this issue in order to diminish these cases of violence.

-In point 2.3 it should say “Data analysis” instead of “Data sources”.

Therefore, I would recommend this paper for publication with minor revisions whether authors can attend to these suggestions and improve the scope of the issue.

Author Response

Response to Reviewer 2 Comments

Point 1: This is a paper with a very interesting topic of study: the different types of violence and its prevalence. It emphasizes how important is to be aware of the scope of this problem and how it influences on different populations, especially children. I think it is relevant as I believe the more information we get about a problematic the more possibilities we have to tackle it. Besides, the authors get interesting results with clear differences according to regions in the types of more common violence.

Response 1: Ok, thanks.

Point 2: Nevertheless it seems more a sociological study than a psychological study. It talks about prevalence but the paper doesn’t mention if the authors analyze the characteristics of all regions to explain the differences and its relevance.

Response 2: This is an ecological study that aims to determine regional prevalence rates, average prevalence by type of aggressor, and provincial spatial clusters for each type of school violence reported in Peruvian educational settings during 2019. We did not have access to reliable information about characteristics of all regions studied, therefore we could not explore the association of these potential characteristics in our analyses. We have included the following sentence about this in the limitations of the study: “Another limitation, of the study was that we did not have access to reliable information about several potential differences between regions to explore the results of our study. Further research is needed.” See line 320-323.

Point 3: Moreover, the paper doesn’t deep on possible measures and tools to address the issue. Authors point to the variables that could increase the prevalence of violence as school size, school climate, disorder and school safety, etc. But they don’t delve into justification to its full extent.

Response 3: We have added the following paragraph: “For example, preventive measures to reduce school violence may include social and emotional learning interventions to improve school climate or to reduce blind spots in the school premises increasing adult supervions to improve school safety. These measures have been used in several evidence-based programs such as Olweus Program (58) or KiVa program (59)” See line 301-305.

Point 4: Furthermore, regarding the sample I find it difficult to assure reliability with the sample and cases of violence.

Response 4: The study used all the cases reported in 2019 within the SíseVe platform of the Ministry of Education. We also cited the decree published in 2018 on the mandatory use of SíseVe in educational institutions to report violence incidents, which is described in the article. See line 318-320. However, we understand that the report of violence incidents may be underrepresented because students may not trust in the system, or they may have worries about the use of the information. We have included this potential report bias in the Discussion: “furthermore, there may be a limitation in the representation of the sample and potential report bias because students may underreport the violence incidents if they fear consequences of providing this kind of information, or cultural stigma of providing sensible data”. See line 315-318.

Point 5: In this sense, authors recognize that the sample could be bias because of the cases in rural areas, with more difficulties to access to the platform to report the information about their situation. Nevertheless, they don’t explain to what extent they can ensure they got most of the cases of violence and how they controlled the process.

Response 5: According to Law 29719, the reporting of school violence cases began in 2014 in SíseVe (see line 293-295). We also cite the decree published in 2018 on the mandatory use of SíseVe in educational institutions to report violence incidents, which is described in the article. (See line 318-320). Although we could not control the process, we hope that our study helps to policy makers in Perú to increase awareness and use of this platform in the schools.

Point 6: Finally, in the conclusions authors only focus on the data they got but they didn’t deep on measures to approach this issue in order to diminish these cases of violence.

Response 6: We have included more discussion about these issues in the text. See lines 336-341.

Point 7: In point 2.3 it should say “Data analysis” instead of “Data sources”.

Response 7: We edited and corrected the name.

Point 8: Therefore, I would recommend this paper for publication with minor revisions whether authors can attend to these suggestions and improve the scope of the issue.

Response 8: We are grateful for the reviewer's suggestions, which helped to improve our paper.
